# Knowledge and Perceptions of Final-Year Nursing Students Regarding Antimicrobials, Antimicrobial Resistance, and Antimicrobial Stewardship in South Africa: Findings and Implications to Reduce Resistance

**DOI:** 10.3390/antibiotics12121742

**Published:** 2023-12-16

**Authors:** Elisma Teague, Selente Bezuidenhout, Johanna C. Meyer, Brian Godman, Deirdré Engler

**Affiliations:** 1Department of Clinical Pharmacy, School of Pharmacy, Sefako Makgatho Health Sciences University, Molotlegi Street, Ga-Rankuwa Zone 1, Ga-Rankuwa 0208, South Africa; elismateague@gmail.com; 2Department of Public Health Pharmacy and Management, School of Pharmacy, Sefako Makgatho Health Sciences University, Molotlegi Street, Ga-Rankuwa Zone 1, Ga-Rankuwa 0208, South Africa; hannelie.meyer@smu.ac.za (J.C.M.); 3South African Vaccination and Immunisation Centre, Sefako Makgatho Health Sciences University, Molotlegi Street, Ga-Rankuwa 0208, South Africa; 4Strathclyde Institute of Pharmacy and Biomedical Sciences, University of Strathclyde, Glasgow G4 0RE, UK; 5Centre of Medical and Bio-Allied Health Sciences Research, Ajman University, Ajman P.O. Box 346, United Arab Emirates

**Keywords:** nurses, knowledge, perceptions, antimicrobials, antimicrobial resistance, antimicrobial stewardship, South Africa

## Abstract

Antimicrobial resistance (AMR) is being increasingly seen as the next pandemic due to high morbidity and mortality rates, with Sub-Saharan Africa currently having the highest mortality rates driven by high rates of inappropriate prescribing in ambulatory care. In South Africa, nurses typically provide a range of services, including prescribing, in public ambulatory care clinics. However, little is currently known about the perception of final-year nursing students regarding antibiotic use, AMR, and antimicrobial stewardship (AMS). Consequently, we sought to address this important evidence gap. A quantitative descriptive study using a self-administered online questionnaire via Google Forms^®^ was undertaken among six universities in South Africa offering a Baccalaureus of Nursing. Knowledge on the classes of antibiotics, organisms covered, and mechanism of action was lacking. The sample size to achieve a confidence interval of 95% with a 5% error margin was 174, increased to 200 to compensate for possible attrition. Only 15.3% of nurses knew that ceftazidime is not a fourth-generation cephalosporin, and only 16.1% knew that clavulanic acid does not decrease inflammation at the site of infection. In addition, only 58.9% and 67.7% agreed that the prescribing of broad-spectrum antibiotics and poor infection control, respectively, increase AMR. AMS was also not a well-known concept among final-year nurses. The lack of knowledge regarding antibiotics, AMR, and AMS among final-year nurses could have important repercussions in practice once these nurses are qualified. Consequently, this information gap needs to be urgently addressed going forward with updated curricula and post-qualification educational activities to reduce AMR in South Africa

## 1. Introduction

Antimicrobial resistance (AMR) is a growing concern globally due to rising rates having their impact on morbidity, mortality, and costs [1,2,3,4,5]. As a result, this phenomenon has been named one of the top ten threats to public health in the world [6] and is increasingly being seen as the next pandemic [7]. In 2019, it was estimated that there were 1.27 million deaths globally directly attributable to bacterial AMR, with possibly up to 4.95 million deaths associated with bacterial AMR [1]. These rates, alongside associated costs, are expected to grow considerably unless addressed [8,9]. The World Bank recently estimated that the costs of AMR could reach as high as USD 3.4 trillion annually unless addressed, which is equivalent to 3.8% of annual global gross domestic product [10]. As a result of the growing concerns, a range of international and national initiatives have now been put in place to try and reduce the rates of AMR and their implications. Initiatives include the World Health Organization’s (WHO) Global Action Plan (GAP) [11], which led to the development of National Action Plans (NAPs) to reduce AMR [12,13].

NAPs are particularly important in lower-income countries, including African countries, with six of the top ten causes of death in these countries being communicable diseases, making AMR a particularly serious problem for some of the poorest countries in the world [1]. African countries, however, are at different stages regarding their introduction and monitoring of activities against the agreed goals in their NAPs [14]. Some of the identified challenges with implementing NAPs across sub-Saharan Africa include knowledge and experience among healthcare professionals (HCPs), quality of the available antimicrobials, grossly inadequate healthcare facilities, limited resources such as access to diagnostics as well as poor surveillance of resistance patterns, and the limited introduction of antimicrobial stewardship programs (ASPs) to drive forward the agreed activities [6,13,14,15,16,17,18]. Consequently, it is anticipated that the economic impact of AMR will be greatest in developing countries and will further increase the economic inequality between countries if not addressed [19]. These multiple challenges need to be addressed going forward given the urgent need to reduce AMR in Africa [1,14,20].

Having said this, there are a number of initiatives and activities ongoing across Africa to improve antibiotic utilisation and reduce AMR. These include initiatives by the African CDC, the African Society for Laboratory Medicine (ASLM), and the Southern Africa Centre for Infectious Disease Surveillance to improve the ongoing surveillance of infectious diseases as well as antibiotic resistance patterns, alongside developing guidelines for African patients to treat common bacterial infections [21,22,23,24,25]. Another recent initiative is the move by the WHO to reclassify antibiotics into their AWaRe classification (Access, Watch, and Reserve), with emphasis on reducing the utilisation of ‘Watch’ and ‘Reserve’ antibiotics with their greater resistance potential [26,27,28]. Alongside this, the WHO AWaRe antibiotic book has recently been launched, which includes treatment suggestions for 26 common or severe clinical syndromes [29,30]. The objective behind the development and introduction of the AWaRe book and guidance is to reduce the inappropriate prescribing of antibiotics, including for self-limiting conditions such as acute respiratory illnesses, thereby reducing AMR [28,29,30].

Other activities promoted in the GAP and NAPs to reduce AMR include the instigation of ASPs, with antimicrobial stewardship (AMS) seen as a comprehensive group of actions encouraging responsible antimicrobial use, thereby reducing AMR and the associated costs [31,32,33]. This is important as the occurrence of AMR is no longer a clinical problem primarily seen in hospital settings as resistant organisms are increasingly detected in patients in primary care [34,35]. However, there have been concerns with the implementation of ASPs in low- and middle-income countries (LMICs), including African countries, due to a lack of both human resources and available finances [36]. This though is changing, and there are now an increasing number of ASPs being introduced in both hospitals and ambulatory care across Africa to improve future prescribing [31,37,38,39,40,41,42].

With respect to the introduction of NAPs across Africa, South Africa does appear to be further ahead than other African countries regarding implementing and monitoring its NAP [14]. This is illustrated by several activities that have already been instigated across South Africa to improve antibiotic prescribing, especially in ambulatory care (Table 1).

Despite these activities (Table 1), there are continuing concerns with rising AMR rates in South Africa [14,62]. These concerns are enhanced by high levels of inappropriate prescribing and dispensing of antibiotics in ambulatory care, with ambulatory care accounting for up to 95% of antibiotic utilisation in LMICs, including among African countries [29,42,52,53,63,64,65,66]. However, this is not always the case, with high rates of compliance to antibiotic treatment guidelines in some studies in South Africa [51].

Currently in South Africa, the public healthcare system dominates providing care for approximately 80% of the population [67]. Within the public system, healthcare services are free at the point-of-care to patients, including medicines, if patients visit public healthcare facilities as opposed to visiting private healthcare professionals (HCPs) or obtaining their medicines directly from community pharmacies [42,67]. These include primary healthcare clinics (PHCs) and community healthcare centres (CHCs) within the primary healthcare system, with currently over 3500 CHCs and PHCs in South Africa [42]. These facilities should be available within 5 km of the residency of over 90% of citizens in South Africa, as well as be free of charge to patients at the point-of-care [53,68]. PHCs are generally smaller than CHCs, with patients typically being seen by nurses rather than by physicians [69,70]. CHCs, on the other hand, are typically larger than PHCs, although less frequent, and out of the two they are the most visited healthcare facility in the public ambulatory care system in South Africa [42]. The principal function of CHCs is to deliver most ambulatory care services to the citizens of South Africa free at the point-of-care. The services typically provided by CHCs include advice on hygiene, vaccinations, and health education, as well as antenatal care in addition to treating patients. Services also include performing examinations as well as referring patients for more specialised care if necessary [51,71]. Whilst physicians are more likely to be present in CHCs than PHCs, nurses do play an appreciable role in managing patients in the public healthcare system in South Africa when they visit ambulatory care facilities [42].

Overall, in South Africa, the nursing profession makes up the largest group of HCPs providing their services at PHCs and CHCs [42,45]. Nurses working in the public ambulatory care system, particularly PHCs, are typically at the frontline of antibiotic prescribing, with prescribing advice typically taken from the Practical Approach to Care Kit (PACK) for adult interventions, which is a symptom-based set of guidelines for managing common conditions in primary care [46,72,73,74]. This is certainly the case for infectious diseases [75]. Consequently, nurses need to be well trained and equipped with the necessary knowledge to prescribe and use antimicrobials appropriately, which is not always the case [52,53,76].

However, to date, little is known about the knowledge and perceptions of nursing students soon to enter the workforce concerning antibiotics, AMR, and AMS. This is important, especially as they will become the principal prescribers of antibiotics in the public ambulatory care system once qualified as South Africa seeks to reduce AMR under its NAP. There have been concerns with the knowledge of nurses in training, along with qualified nurses, regarding antibiotics, AMR, and ASPs in other countries, including other LMICs [77,78,79,80,81,82]; however, this is not always the case [83]. There have also been concerns with the knowledge, attitude, and practices (KAPs) towards antibiotics, AMR, and AMS among practicing nurses in South Africa, with requests for additional education and training to help with prescribing [61]. In addition, there have been concerns with adherence to standard treatment guidelines (STGs) among nurses in South Africa [52,53,84]; however, this is again not necessarily the case in all instances [51,85]. There is also variation across LMICs with regard to the extent to which education pertaining to areas such as AMS are currently being incorporated into nursing students’ curricula [86]. Consequently, this study was undertaken to address this information gap in South Africa, with the findings serving as future guidance to all key stakeholder groups.

## 2. Results

### 2.1. Response Rate

Out of our target universities, one university declined participation, resulting in a total of five universities taking part in the survey. A total of 242 final-year students were enrolled for the Baccalaureus of Nursing (BCur) at the five institutions, with 124 finally participating, giving a response rate of 51.2% (n = 124).

### 2.2. Knowledge and Education on Antimicrobials

Most of the student nurse participants (77.4%) were aware that antimicrobials include antibiotics, antivirals, antifungals, and antiparasitics; however, almost 20% of the final-year nurse participants thought antimicrobials only applied to antibacterials.

The majority (97.58%; n = 121) agreed that good knowledge of antimicrobials is important for nursing professionals (Table 2). Three quarters (74.2%; n = 92) of the student nurse participants believed their education on antimicrobials was sufficient, and 64.5% (n = 80) felt equipped to select the appropriate antimicrobial regimen to treat presenting infectious diseases when qualified.

There were concerns with some of the general pharmacologic knowledge regarding antimicrobials (Table 3). Nearly 20% of the final-year student nurse participants either agreed (8.9%; n = 11) or were unsure (9.7%; n = 12) whether aspirin is an antibiotic.

There were also concerns with the final-year nurses’ knowledge of ceftazidime, despite this being included in the current South African Standard Treatment Guidelines and Essential Medicine List (STG&EML), and generally seen to be a good antibiotic choice to cover Gram-negative organisms. Only 15.3% of final-year nurses correctly disagreed with the statement that ceftazidime is a fourth-generation cephalosporin, while 50.8% (n = 63) were unsure. In total, 52.4% (n = 65) incorrectly agreed that ceftazidime is a good choice to treat Gram-positive infections, while 40.3% (n = 50) were unsure.

Alongside this, 43.6% (n = 54) believed that antibiotics are indicated to reduce any kind of pain and inflammation, with 12.9% being unsure. In addition, 33.1% (n = 41) believed that clavulanic acid is given with amoxicillin (co-amoxiclav) to decrease inflammation at the site of infection, with 50.8% being unsure (Table 3).

However, 58.1% (n = 72) of final-year nurses correctly identified erythromycin as a macrolide, and 77.4% (n = 96) agreed that antibiotics cannot be used to treat human papilloma virus (HPV). Concurrent with this, only 18.5% (n = 23) agreed that antibiotics are useful for treating cold and influenza symptoms, and only 9.7% (n = 12) believed that antibiotics are useful for treating viral infections. In addition, only 9.7% of the future nurses agreed that patients may stop the use of antibiotics as soon as they feel better (Table 3).

In addition, 55.6% (n = 69) did not agree to the simultaneous use of two types of antibiotics. Overall, several of the answers indicated a lack of pharmacology knowledge and understanding regarding antibiotics and AMR.

### 2.3. Perceptions of Antimicrobial Resistance

The majority (87.9%; n = 109) of future nurses considered their role in curbing AMR as crucial; however, 21.8% (n = 27) were not aware that this phenomenon poses a global threat. Alongside this, whilst 82.3% (n = 102) of future nurses believed they had received sufficient education pertaining to AMR, there were concerns with a lack of knowledge and understanding regarding AMR in a number of areas (Table 4). This is illustrated by only 34.7% (n = 43) of future nurses agreeing that the overuse of antibiotics is a principal risk factor for AMR, only 12.9% disagreeing that the prescribing of broad-spectrum antibiotics increases AMR, and 55.6% (n = 69) being unsure that exposure to antibiotics appears to be the principal risk factor increasing AMR (Table 4). In addition, only 22.6% (n = 28) knew one of the mechanisms for acquiring resistance, with the majority being unsure (67.74%; n = 84).

However, 83.1% (n = 103) agreed that AMR can be minimised by using narrow-spectrum antibiotics after the identification of potential organisms, and 84.7% (n = 105) believed that AMR can be minimised by improving bacterial diagnostics (Table 4).

### 2.4. Knowledge and Perceptions of Antimicrobial Stewardship

Of concern is that 66.9% (n = 83) and 63.7% (n = 79) of future nurses, respectively, had never heard of AMS nor ASPs. A mere 3.2% (n = 4) heard about AMS at university, while another 12.9% (n = 16) came to know about AMS activities while at university and working as a student nurse in practice.

More than 80% (82.26%, n = 102) were not aware of any ASPs in South Africa.

## 3. Discussion

We believe that this is one of the first studies conducted in South Africa assessing the KAPs of final-year nursing students towards antibiotics, AMR, and ASPs. This builds on studies, including Balliram et al.’s study (2021), where there was generally good knowledge regarding antibiotics and viral infections; however, there was variable confidence among nurses regarding the prescribing of antibiotics [61]. In addition, in their study, Engler et al. (2021) found that many HCPs working in public facilities, including nurses, could not recall receiving any training on antimicrobials and AMR during their education [54].

The response rate in our study was favourable, at 51.2%, with most studies aiming for a 55% to 60% response rate. However, response rates to web-based surveys are highly variable and traditionally in the range of 25–30% [87].

In our study, there was variable knowledge among final-year nursing students in South Africa, including knowledge of AMR and ASPs, which is similar to studies involving student nurses and qualified nurses in other countries [82,88,89,90,91], as well as similar to the review by Fuller et al. (2023) involving nurses, HCPs, and others across Africa [92].

Despite 74.2% of the final-year nurses in our study believing that their education on antimicrobials was sufficient, and 64.5% feeling equipped to select the appropriate antimicrobial regimen to treat presenting infectious diseases, there were concerns with some of their knowledge. Nearly 20% of final-year nurses either agreed or were unsure whether aspirin is an antibiotic. However, this is lower than 57.7% of healthcare students in Saudi Arabia, who were mainly nurses, believing that paracetamol is an antibiotic [93]. In addition, up to 66.7% of trainee nurses in Spain believed that aspirin is an antibiotic, although this figure was appreciably lower among final-year nurse students (16.7%) [79].

There were also concerns with final-year nurses’ knowledge of ceftazidime, despite this being included in the current South African STG&EML and generally seen to be a good choice of antibiotic to cover Gram-negative organisms [94]. As ascertained, only 15.3% of the final-year nurses in our study correctly disagreed about ceftazidime being a fourth-generation cephalosporin, while 50.8% were unsure. A total of 52.4% also incorrectly agreed that ceftazidime is a good choice for Gram-positive infections, while 40.3% were unsure. This is comparable with a study conducted in Sri Lanka, where despite 91.5% of trainee nurses stating that antibiotics are active against bacteria, only 72.4% of the trainee nurses surveyed had any awareness about the differences in the spectrum activity of the various antibiotics in the questionnaire [82].

However, encouragingly, only 18.5% of the final-year nurses in our study agreed that antibiotics are useful for treating cold and influenza symptoms, 9.7% agreed that antibiotics are useful for treating viral infections, and 9.7% agreed that patients may stop the use of antibiotics as soon as they feel better. This is comparable with trainee nurses in Sri Lanka, where 41.2% believed that antibiotics are effective against the common cold, 22.1% stated that they would stop their antibiotics after a rapid recovery, and 27.1% stated that they would save the remaining antibiotics for another infectious disease episode [82]. Similarly, another study in Sri Lanka involving healthcare students, including nursing students, found that 46% believed that antibiotics are appropriate for treating a sore throat [83]. In addition, up to 46.2% of trainee nurses in Spain believed that antibiotics are useful for viral infections; however, again, this was lower among final-year student nurses (19.2%) [79]. Having said this, no student in Spain agreed that antibiotics should be prescribed in patients with coughs and colds to aid recovery [79]. There were also concerns among nurses in Saudi Arabia, where only 44.4% agreed that treating a viral infection with an antibiotic is inappropriate [95].

Furthermore, there were concerns with knowledge of antibiotics among 54.7% of nursing students in Ghana [88]. Similarly, in Iraq, 55.5% had poor knowledge regarding the rational use of antibiotics [96]. There were also concerns with nurses’ knowledge of antibiotics in Saudi Arabia, where 54.3% had only moderate knowledge, with 19.4% having poor knowledge [95].

Infection prevention is seen as one of the most effective methods to slow the spread and development of AMR [97,98]. However, 20.2% of the final-year nurses in our study disagreed with this, and another 12.1% were unsure. This, however, compares favourably with Spain, where nursing students typically showed a lack of knowledge in terms of AMR [79]. There were also concerns with the limited involvement of nurses in ASPs in Pakistan; however, this was despite positive attitudes and knowledge towards ASPs [77,89]. In Scotland, knowledge of AMR and AMS increased significantly among nursing students following educational input [86]. This is encouraging as 84.7% and 83.1% of the final-year nurses, respectively, in our study agreed that AMR can be minimised by improving bacterial diagnostics as well as by de-escalation after identification and susceptibility testing of the pathogen.

Overall, it is of fundamental importance to improve the knowledge and understanding of nursing students in South Africa and beyond regarding antibiotics, AMR, and ASPs. This is because, as mentioned, nurses in South Africa are the largest category of healthcare providers in ambulatory care in the public sector [99], and any pertinent knowledge gaps need to be urgently identified and addressed in the undergraduate curriculum and continued post-qualification as part of continuous professional development (CPD) activities. Consequently, in South Africa, during their education, future nurses should be made fully aware of the Antimicrobial Resistance National Strategy Framework in South Africa and embed with the principles of AMS [46,100]. This would raise their awareness of the current challenges surrounding AMR in South Africa and reinforce the crucial role that nurses play in enhancing AMS as well as infection prevention and control measures across the sectors, especially ambulatory care. We are beginning to see more ASPs being introduced in South Africa across the sectors [37,42,55,56,101], and nurses should be an increasing part of these activities going forward, especially given their critical role in ambulatory care. Alongside this, student nurses in South Africa need to be aware of the WHO’s AWaRe classification for antibiotics, especially reducing the prescribing of ‘Watch’ and ‘Reserve’ antibiotics with their increasing resistance potential [26,28]. The recently launched WHO AWaRe book containing treatment suggestions for infections typically seen in ambulatory care should help here [29,30], and the guidelines must be a key element of the nursing curriculum going forward.

Pharmacists could also augment the knowledge of fellow HCPs, including nurses, by promoting the appropriate use of antimicrobials, thereby enriching, and imbedding the knowledge of nurses when working with them to improve future antibiotic prescribing [31,102,103,104]. We will continue to monitor these developments.

We are aware of the limitations of this study. Firstly, we do recognise that the study design did not allow for the causal investigation of variables. In addition, we only included nurses from one province in South Africa, and we only targeted final-year nursing students. However, we believe a strength of this study is that only one of the six universities providing the BCur qualification in the Gauteng province did not participate in this study. Consequently, despite these limitations, we believe that our findings are robust, providing direction for the future as South Africa struggles to reduce AMR rates in line with the goals of the NAP.

## 4. Materials and Methods

After ethical approval was granted by the Sefako Makgatho University Research Ethical Committee (SMUREC) (SMUREC/P/130/2020: PG), a quantitative descriptive study design was utilised to meet the objectives of this study.

### 4.1. Study Population and Sample Size

Permission to conduct the study was obtained from each of the six institutions offering the BCur qualification in Gauteng. The target population included final-year students registered for and studying BCur at universities across Gauteng. All the universities in Gauteng offering the BCur degree were included in the study. i According to the South African Nursing Council (SANC), three of the six universities were accredited to provide the new curriculum [99,105], with the number of the annual intake of students in these three universities contained within Table 5.

The other three institutions were contacted directly via e-mail to confirm the number of students at 24, 34 and 70 respectively. The target population, therefore, included 298 final-year students studying BCur at the six universities in Gauteng Province. The required sample size (n), when conducting the study at a 95% confidence interval with a 5% margin of error, assuming a 50% response distribution, was 174. To compensate for attrition and incomplete/invalid forms, the target sample size was increased to 200.

### 4.2. Data Collection Instrument and Procedures

Data were collected from final-year nursing students using a self-administered online questionnaire, adapted from previous studies [79,81,82]. We, together with other researchers, have used this approach before when conducting similar studies [63,88,98,107].

The online questionnaire was pre-tested by nine academic pharmacist interns from one of the institutions who were not part of the study population. The link to Google Forms^®^ worked optimally and was linked to the researchers’ email addresses. The average time to complete the questionnaire was eleven minutes. All the comments and suggestions raised were considered and the questions amended accordingly before full roll out.

A participant could only continue with the questionnaire after providing informed consent, could only complete the questionnaire once, and every question was marked as ‘required’ to ensure all the questions were answered.

The electronic questionnaire contained sections on antimicrobials (22 questions), AMR (19 questions), and AMS and ASPs (six questions) using a 3-point category scale, as well as a 5-point Likert scale in a multiple-choice format [108,109,110]. The 5-point Likert scale options ranged from ‘Strongly disagree’, ‘Disagree’, ‘Not sure’, ‘Agree’ to ‘Strongly agree’. For analysis and interpretation, the responses to all the questions were subsequently collapsed into three categories, i.e. Disagree, Not sure, and Agree, where relevant. To assess the knowledge of the participants, a set of designed questions was asked regarding antimicrobials, AMR, and AMS.

Key questions tested general and specific knowledge and awareness of antibiotics, e.g., the term “antimicrobial”, the use of antibiotics to treat flu symptoms and viral infections, the different classifications, and their indications. The latter included ceftazidime, a 3rd-generation cephalosporin, which is included in the South African STG/EML and considered to be a good choice to cover Gram-negative organisms. Questions regarding factors contributing to AMR and interventions to restrain this phenomenon were included, as well as their perceptions about training received on these matters (questionnaire included in the Appendix A).

Typically, general principles applied across different groups of microorganisms, with bacteria and viruses causing the most infections, followed by fungi, protozoa, and helminths [97,111]. Antimicrobials to treat such infections are seen as inclusive of antibiotics, antivirals, antifungals, and antiparasitics [112].

Data were collected from each institution over a period of 12 weeks. An email containing the link to the online questionnaire was sent to the various heads of department (HODs) of the participating institutions for distribution to their students. The HODs were “blind copied” and could not use the “reply all” function, thus ensuring confidentiality. Reminder emails were sent fortnightly, or the HODs were contacted telephonically, in case of slow responses. The overall data collection period was from 5 February 2021 to 31 July 2021. The data collected were available only to the principal researcher (ET) and supervisors. Participating institutions were allocated a unique identifier code, e.g., Uni-1, Uni-2, etc., to maintain anonymity. Completed Google Forms^®^ were exported to Microsoft Office Excel^®^ and the data were checked for possible errors or duplication before statistical analyses were performed using SPSS statistics for Windows, version 25 (IBM Corp., Armonk, NY, USA).

We deliberately did not exclude studies involving nurses from higher-income countries when comparing our findings with other countries, unlike previous studies that some of the authors have been involved with [113,114]. This is because we believed that key stakeholders in South Africa could potentially learn from higher-income countries if this was the case. Alternatively, we feel that other countries have similar issues that need addressing whatever their income level, and South Africa is not alone in this respect.

## 5. Conclusions

Overall, the knowledge and understanding of final-year nursing students in South Africa regarding antibiotics, AMR, and ASPs were concerning. AMR is now a public health priority; however, the student nurses were typically unaware of AMR being an increasing global threat brought about by the overuse and inappropriate use of antimicrobials. Overall, the level of knowledge that student nurses receive on antibiotics, AMR, and AMS appears inadequate considering the crucial role they will play in the prescribing of antibiotics, particularly in ambulatory care, post-qualification. This needs to be urgently addressed going forward, including CPD activities, if the goals of the NAP to reduce AMR in South Africa are to be attained.

## Figures and Tables

**Table 1 antibiotics-12-01742-t001:** Ongoing activities by key national groups in South Africa to improve antibiotic prescribing, including ambulatory care.

Activity	Reference
National Action Plan/Antimicrobial Resistance National Strategy Framework 2014–2024; regular monitoring of its implementation with updates together with active surveillance of AMR	[14,43,44,45,46]
Developing and broadcasting a national manual to improve infection prevention and control across all sectors of care in South Africa	[47]
Updating the Standard Treatment Guidelines/Essential Medicine List (STG/EML), including recommendations for the management of COVID-19 as well as the general management of infections in ambulatory care, including UTIs	[48,49,50]
Assessment and monitoring of the prescribing of antibiotics in ambulatory care vs. recommendations in the STG/EML	[42,51,52,53]
Encouraging and assessing antimicrobial stewardship activities across the sectors, including implementing ASPs among public healthcare facilities in South Africa	[42,54,55,56]
Encouraging South African citizens to become antibiotic guardians	[57]
Refining the curricula of student healthcare professionals, including nurses, to improve their knowledge regarding antibiotics, AMR, and ASPs alongside updating continuous professional development activities post-qualification to address pertinent knowledge and training gaps. On the other hand, in one recent qualitative study, surveyed healthcare professionals could not recall receiving any antimicrobial training, including at the undergraduate level [54].	[58,59,60,61]

AMR = antimicrobial resistance; ASPs = Antimicrobial Stewardship Programme; UTIs = urinary tract infections.

**Table 2 antibiotics-12-01742-t002:** Opinion regarding own knowledge and education on antimicrobials (n = 124).

Statements Posed to Participants	Response; n (%)	
A strong knowledge of antimicrobials is important in my career.	Agree	121 (97.6%)
	Not sure	1 (0.8%)
	Disagree	2 (1.6%)
I have received sufficient education in pharmacology to select the best antibiotic(s) for a specific infection.	Agree	92 (74.2%)
	Not sure	19 (15.3%)
	Disagree	13 (10.5%)
I have received sufficient education to select an appropriate regimen (dose, route, frequency) of antibiotic therapy.	Agree	80 (64.5%)
	Not sure	33 (26.6%)
	Disagree	11 (8.9%)

**Table 3 antibiotics-12-01742-t003:** General and specific knowledge questions regarding antimicrobials (n = 124).

Statements Posed to Participants	Response; n (%)		
	Agree	Not Sure	Disagree
**Aspirin is an antibiotic.**	11 (8.9%)	12 (9.7%)	**101 (81.4%)**
**Ceftazidime is a fourth-generation cephalosporin antibiotic.**	42 (33.9%)	63 (50.8%)	**19 (15.3%)**
**Ceftazidime is a good choice to cover Gram-positive organisms.**	65 (52.4%)	50 (40.3%)	**9 (7.3%)**
**Antibiotics are used to treat cold and flu symptoms.**	23 (18.5%)	9 (7.3%)	**92 (74.2%)**
**Antibiotics are useful in treating viral infections.**	12 (9.7%)	12 (9.7%)	**100 (80.6%)**
**Antibiotics are indicated to reduce any kind of pain and inflammation.**	54 (43.6%)	16 (12.9%)	**54 (43.5%)**
**Antibiotics can cause secondary infections after killing good bacteria present in our body.**	64 (51.6%)	28 (22.6%)	**32 (25.8%)**
**Erythromycin is a macrolide antibiotic.**	**72 (58.1%)**	7 (5.6%)	45 (36.3%)
**Antibiotics can cause allergic reactions.**	**116 (93.6%)**	5 (4.0%)	3 (2.4%)
**Patients may stop the use of antibiotics as soon as they feel better.**	12 (9.7%)	7 (5.6%)	**105 (84.7%)**
**Two different types of antibiotics may not be prescribed for simultaneous use.**	39 (31.5%)	16 (12.9%)	**69 (55.6%)**
**Antibiotics should always be prescribed as prophylaxis to prevent future infections.**	57 (45.97%)	10 (8.08%)	**57 (45.97%)**
**Antibiotics cannot treat Human Papilloma Virus (HPV).**	**96 (77.4%)**	11 (8.9%)	17 (13.7%)
**Clavulanic acid is given with amoxicillin (AMOXICLAV) to decrease inflammation at the site of infection.**	41 (33.1%)	63 (50.8%)	**20 (16.1%)**
**Inappropriate use of antimicrobials can harm patients.**	**106 (85.5%)**	11 (8.9%)	7 (5.6%)

Correct answers are highlighted in bold.

**Table 4 antibiotics-12-01742-t004:** Perceptions regarding antimicrobial resistance (n = 124).

Statements Posed to Participants	Response; n (%)		
	Agree	Not Sure	Disagree
**Prescribing a broad-spectrum antibiotic increases antibiotic resistance.**	**73 (58.9%)**	35 (28.2%)	16 (12.9%)
**Poor infection control practices by healthcare professionals cause the spread of antibiotic resistance.**	**84 (67.7%)**	15 (12.1%)	25 (20.2%)
**Exposure to antibiotics appears to be the principal risk factor for the appearance of antibiotic-resistant bacteria.**	**43 (34.7%)**	69 (55.6%)	12 (9.7%)
**Antibiotic resistance can be minimised by using narrow-spectrum therapy after identification and susceptibility testing of infectious bacteria.**	**103 (83.1%)**	18 (14.52%)	3 (2.42%)
**Bacteria may acquire efflux pumps that extrude the antibiotic from the cell.**	**28 (22.6%)**	84 (67.7%)	12 (9.7%)
**Improving bacterial diagnostics will allow combating antibiotic resistance.**	**105** **(84.7%)**	13 (10.5%)	6 (4.8%)

Correct answers are highlighted in bold.

**Table 5 antibiotics-12-01742-t005:** Number of student intake of SANC-accredited universities in Gauteng (adapted from [106]).

Accredited University in Gauteng Offering Bachelor of Nursing	Number of Students per Intake
University A	Sixty (60)
University B	Sixty (60)
University C	Fifty (50)

## Data Availability

Additional data are available on reasonable request from the corresponding authors.

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
