# Peer review of "Knowledge and Perceptions of Final-Year Nursing Students Regarding Antimicrobials, Antimicrobial Resistance, and Antimicrobial Stewardship in South Africa: Findings and Implications to Reduce Resistance"

_antibiotics, 2023, doi:10.3390/antibiotics12121742_

Round 1
Reviewer 1 Report
Comments and Suggestions for Authors
Abstract. Does not state sample size or survey method.
Introduction:
Explains importance;
Line 91: ‘we are seeing’ – unclear who is ‘we’ omit
Line 109: ‘accounting for’ – replace with ‘providing treatment / care for
Line 113: ‘these’ – unclear whether refer to community pharmacies or public health care facilities
Line 153 Results – Section on methods is missing - how was survey administered ?
Line 154: Response rate – method of accessing respondents not provided;
Line 156: What is BCur ?
Line 163: ‘is’ present tense, while other sentences use past tense
Line 176 : what is “STG and EML”
Line 209: ‘acquiring resistance’ - add ‘for’ acquiring resistance
Line 217: omit ‘subsection’ wording
Line 218 : ‘of concern’ – singular not plural
Line 241: the comparison with studies in other countries would need to take into account the method of administration, stage of training etc I’m not sure that this paragraph adds a great deal, as the contexts in other countries may vary
Line 252ff : again the comparison with other countries needs to take into account factors such as the stage of training, method of administration etc. Limiting the comparison to other African or LMIC countries may be more useful.
Line 281: this is a useful result, and should be included in the result section, rather than introduced in the discussion section
Line 315: Materials and methods. Traditionally this section precedes the results section, and assists the reader in understanding the results. Suggest move to precede results.
Methods does not refer to ethical review – but noted in the supplementary materials section.
Supplementary materials
Letter and explanation is useful, but not necessary to include the questionnaire.
Comments on the Quality of English LanguageSee comments above
Author Response
Comments and Suggestions for Authors:
Author comments: Thank you for your comments. We hope we adequately address these
1) Abstract. Does not state sample size or survey method.
Author comments: The following was added: A quantitative descriptive study using a self-administered online questionnaire via Google FormsÒ was undertaken among six universities offering a Baccalaureus of Nursing. Knowledge on the classes of antibiotics, organisms covered, and mechanism of action was lacking. The sample size to achieve a confidence interval of 95% with a 5% error margin was 174, increased to 200 to compensate for attrition. Pertinent details have now been added in, and we trust this is now OK.
2) Introduction: Explains importance;
Author comments: Thank you for this – appreciated!
3) Line 91: ‘we are seeing’ – unclear who is ‘we’ omit
Author comments: Thank you – now updated
4) Line 109: ‘accounting for’ – replace with ‘providing treatment / care for
Author comments: Thank you – now updated
5) Line 113: ‘these’ – unclear whether refer to community pharmacies or public health care facilities
Author comments: Thank you – now updated
6) Line 153 Results – Section on methods is missing - how was survey administered ?
Author comments: Thank you for this. In line with the requirements from the Journal – the Methods Section typically appears at the end of the manuscript after Discussion and before the Conclusions. We trust this is OK with you.
7) Line 154: Response rate – method of accessing respondents not provided;
Author comments: Thank you for this. In line with the requirements from the Journal – the Methods Section appears after ‘Discussion’ with these details. We trust this is OK with you.
8) Line 156: What is BCur ?
Author comments: Thank you for this comment. We have now moved the full name to its first appearance (Bottom of Page 4).
9) Line 163: ‘is’ present tense, while other sentences use past tense
Author comments: Thank you – this phrase using ‘is’ was taken from the questionnaire. We hope this is OK with you.
10) Line 176 : what is “STG and EML”
Author comments: Thank you – now updated
11) Line 209: ‘acquiring resistance’ - add ‘for’ acquiring resistance
Author comments: Thank you – now updated
12) Line 217: omit ‘subsection’ wording
Author comments: Thank you – now updated
13) Line 218 : ‘of concern’ – singular not plural
Author comments: Thank you – now updated
14) Line 241: the comparison with studies in other countries would need to take into account the method of administration, stage of training etc I’m not sure that this paragraph adds a great deal, as the contexts in other countries may vary
Author comments: Thank you for this comment. However, in this study we wanted to compare the situation in South Africa with a range of other countries especially developed countries with greater resources to see if they could provide additional direction – especially with rather recent studies. We have now added in further details regarding this at the end of the Methodology section. Interestingly, despite both aspirin and paracetamol being very common drugs, even in high income countries students still classified them as antibiotics. Consequently, nurses in South Africa were not alone! We hope this is now acceptable with you.
15) Line 252: again the comparison with other countries needs to take into account factors such as the stage of training, method of administration etc. Limiting the comparison to other African or LMIC countries may be more useful.
Author comments: Thank you for this. As discussed above, we included higher-income countries in case they could provide guidance, etc., in South Africa (and now inputted into the Methodology Section). However – as seen – still issues in e.g. Saudi-Arabia and Spain along with LMICs such as Sri-Lanka as well as other African countries (Review of Fuller et al – 2023 top of Page 10). This shows that there are a number of issues among nurses across countries regarding antibiotics, AMR, ASPs, etc., – as well as South Africa – that need to be addressed going forward with rising AMR rates. We hope this is now acceptable to you.
16) Line 281: this is a useful result, and should be included in the result section, rather than introduced in the discussion section.
This is encouraging as 84.7% and 83.1% of final year nurses respectively in our study agreed that AMR can be minimised by improving bacterial diagnostics as well as by de-escalation after identification and susceptibility testing of the pathogen.
Author comments: Thank you for this comment – adding to the Discussion, Please note though that these results have been included in the results section - Table 4 and the bottom of Page 8.
17) Line 315: Materials and methods. Traditionally this section precedes the results section, and assists the reader in understanding the results. Suggest move to precede results.
Author comments: Thank you for this. In line with the requirements from the Journal – the Methods Section appears at the end. We trust this is OK with you.
18) Methods does not refer to ethical review – but noted in the supplementary materials section.
Author comments: Thank you – now made this clearer at the start of the Methodology section in addition to the Institutional Review Section just above the References. We trust this is now OK.
19) Supplementary materials - Letter and explanation is useful, but not necessary to include the questionnaire.
Author comments: Thank you for this. We understand this – but would like to keep this if we can.
20) Comments on the Quality of English Language See comments above
Author comments: Thank you for your help – appreciated!

Reviewer 2 Report
Comments and Suggestions for Authors
Comments on the Quality of English LanguageAuthor Response
Comments and Suggestions for Authors - Reviewer comment
A) Introduction
1) I refute this statement: ‘…making AMR a particularly serious problem for some of the poorest countries in the world’ (page 2, lines 61). I think that AMR is threatening worldwide. In the poorest countries, diagnostic stewardship is a limitation and rarely justifies the severity of disease caused by resistant bacteria (underestimate the burden). They not think that AMR is a serious problem.
Reference: Rolfe, R., Kwobah, C., Muro, F. et al. Barriers to implementing antimicrobial stewardship programs in three low- and middle-income country tertiary care settings: findings from a multi-site qualitative study. Antimicrob Resist Infect Control 10, 60 (2021). https://doi.org/10.1186/s13756- 021-00929-4; Gulumbe BH, Haruna UA, Almazan J, Ibrahim IH, Faggo AA, Bazata AY. Combating the menace of antimicrobial resistance in Africa: a review on stewardship, surveillance and diagnostic strategies. Biol Proced Online. 2022 Nov 23;24(1):19. doi: 10.1186/s12575-022-00182-y; Zakhour, J., Haddad, S. F., Kerbage, A., Wertheim, H., Tattevin, P., Voss, A., Ünal, S., Ouedraogo, A. S., Kanj, S. S., & International Society of Antimicrobial Chemotherapy (ISAC) and the Alliance for the Prudent Use of Antibiotics (APUA) (2023). Diagnostic stewardship in infectious diseases: a continuum of antimicrobial stewardship in the fight against antimicrobial resistance. International journal of antimicrobial agents, 62(1), 106816. https://doi.org/10.1016/j.ijantimicag.2023.106816
Author comment: Thank you – we agree and have included a number of these references in the updated paper. However – the Lancet Study which we quote (Ref 1) highlights that the greatest burden of AMR is currently in sub-Saharan Africa and will worsen with variable implementation of the NAPs among the different African countries which we comment on (This research – Ref 14 – was led by one of the co-authors – Prof. Brian Godman – supported by another – Prof Johanna Meyer). Many factors add to this burden including over prescribing of antibiotics in ambulatory care especially for self-limiting conditions such as ARIs – as well as extensive self-purchasing of antibiotics again for essentially self-limiting conditions. Stewardship and increased training of nurses can help here – which we build on. We trust this is now acceptable.
2) There is a new edition of this statement ‘recently launching the WHO AWaRe antibiotic book, which includes treatment suggestions for 26 common or severe clinical syndromes’ (page 2, lines 79). The objective is to reduce inappropriate prescribing of antibiotics, including for self-limiting conditions such as acute respiratory illnesses, thereby reducing AMR [25-27]. (page 2, lines 80). - This statement is ambiguous. Is that the objective of the (authors’) study or the WHO AWaRe?
Author comments: Thank you – now updated. Two of the authors are in fact currently working extensively with the St Geroge’s team (Mike Sharland and colleagues) to update quality/ prescribing indicators based on the AWaRe book to improve prescribing in ambulatory care across countries – especially LMICs, which hopefully will become key indicators in the future.
3) Is there a national policy for nurses prescribing antibiotics in South Africa? (Page 4 line 131 of original manuscript)
Author comment: Thank you for this. For your information. PACK is a system-based set of guidelines and incudes communicable diseases (TB, HIV, STD). We have now updated this including the additional reference. These guidelines together with the Standard Treatment Guidelines/Essential Medicine List should be applied to the Antimicrobial Resistance National Strategy Framework/ Nurses activities to improve future prescribing which we comment on in the updated paper. We trust this is now OK.
B) Discussion
1) What these statements (page 9 lines 303) discuss about (which result)?
Author comment: Apologies, I am not quite sure what has been referred to. However, we believe pharmacists through their training can help guide nurses on the appropriate use of antibiotics – and we have already seen in Africa examples or pharmacist-led ASPs (which we quote). We have now made this clearer in the revised paper and trust this is now OK.
C) Conclusions
1) The respondent of this study are nurses’ students. The author should add recommendation associated with the finding. This conclusion (… CPD activities) is not comes from the results.
Author comment: Thank you, but we do not fully understand this comment. Yes, the respondents were final year nursing students. The results highlighted gaps in their knowledge regarding antimicrobials, AMS and AMR. In the discussion section, the problems/ issues with current knowledge were highlighted, and key recommendations made. We trust this is now acceptable.
